# Parkinson’s Disease, Periodontitis and Patient-Related Outcomes: A Cross-Sectional Study

**DOI:** 10.3390/medicina56080383

**Published:** 2020-07-30

**Authors:** Patrícia Lyra, Vanessa Machado, Luís Proença, Josefa Domingos, Catarina Godinho, José João Mendes, João Botelho

**Affiliations:** 1Clinical Research Unit (CRU), Centro de Investigação Interdisciplinar Egas Moniz (CiiEM), Instituto Universitário Egas Moniz, 2829-511 Caparica, Portugal; patricialyra10@gmail.com (P.L.); vmachado@egasmoniz.edu.pt (V.M.); cgodinho@egasmoniz.edu.pt (C.G.); jmendes@egasmoniz.edu.pt (J.J.M); 2Periodontology Department, Clinical Research Unit (CRU), CiiEM, Egas Moniz, CRL, 2829-511 Caparica, Portugal; 3Quantitative Methods for Health Research Unit (MQIS), CiiEM, Egas Moniz, CRL, 2829-511 Caparica, Portugal; lproenca@egasmoniz.edu.pt; 4Laboratory of Motor Behavior, Sport and Health Department, Faculty of Human Kinetics, University of Lisbon, 1495-751 Lisbon, Portugal; domingosjosefa@gmail.com

**Keywords:** Parkinson’s disease, movement disorders, Parkinsonian disorders, oral health, periodontitis, periodontal diseases, quality of life

## Abstract

*Background and objectives:* People with Parkinson’s disease (PD) may be at risk of having bad periodontal status. A consistent periodontal examination is critical to investigate how it impacts on PD quality of life. We aimed to assess the periodontal status of people with PD, and its association with quality of life and self-perceived xerostomia. *Materials and Methods:* To this end, from February to March 2020, we consecutively enrolled 28 PD individuals, and motor and non-motor symptoms of PD were assessed using the Movement Disorder Society Unified Parkinson’s Disease Rating Scale (MDS-UPDRS). We performed full-mouth periodontal examination and gathered information on self-perceived quality of life in PD, oral health impact profile (OHIP-14) and xerostomia. *Results:* The prevalence of periodontitis was 75.0% and most cases were identified as severe (46.4%). Upper extremity rigidity, hand posture and kinetic tremors were significantly correlated with worse periodontal status. PDQ-8 showed to be correlated with self-perceived oral health-related quality of life and xerostomia levels. *Conclusions:* This group of people with PD had a high prevalence of periodontitis. Deteriorated levels of the upper extremities in advanced stages of PD were associated with worse periodontal status and hygiene habits. Quality of life in PD appears to be associated with self-perceived OHRQoL and xerostomia.

## 1. Introduction

Parkinson’s disease (PD) is one of the most frequent, disabling and progressive neurodegenerative conditions [1,2,3]. It is a growing condition, especially in a globally aged population [4]. PD prevalence increases with age, affecting 1% of individuals over 60 and up to 4% in higher age groups [3]. PD is clinically characterized by a multitude of motor and nonmotor features, heavily affecting patient’s quality of life [5]. PD motor symptoms such as resting tremor, rigidity and bradykinesia are all commonly targeted by the standard available therapies, which include dopaminergic drugs and functional neurosurgery [1,5]. Other symptoms may include loss of balance, gait dysfunction, swallowing and speech impairment, autonomic disturbances and cognitive impairments [5]. PD progression may interfere with daily activities, among them oral hygiene habits, increasing the risk for oral diseases.

Periodontitis is a chronic, polymicrobial and inflammatory disease of the oral cavity, which is characterized by chronic inflamed gums and bone destruction surrounding the teeth due to a dysbiotic microflora [6,7]. Periodontitis is one of the most prevalent diseases, being its severe form the 6th most prevalent condition worldwide [8,9]. The onset and progression are triggered by inadequate oral hygiene behaviours and motor hygiene impairments, becoming more prevalent with age [10,11]. Periodontitis is associated with masticatory dysfunction and impacts negatively on the patient’s oral health-related quality of life (OHRQoL) [12], which can be restored after successful periodontal treatment [13].

Periodontitis has been consistently associated with a number of chronic diseases, such as diabetes mellitus [14], cardiovascular diseases [15,16], neurological diseases such as Alzheimer’s [17], rheumatoid arthritis [18], solid organ transplants [13] and stress [19]. Nevertheless, the oral status in people with PD is poorly studied. A number of observational studies reported weakened oral health status and reduced oral hygiene care in PD patients [20,21,22,23], although the periodontal assessment performed in these studies was inadequate because partial mouth strategies were employed that increases reporting bias risk [13,24]. Furthermore, data from Taiwan’s National Health Insurance Research Database revealed that periodontal inflammatory disease may increase the risk of developing PD [25], though clinical definitions followed the ninth revision of the International Classification of Diseases (ICD-9-CM) and lack scientific robustness. Therefore, assessing the periodontal status in PD patients with a consistent periodontal examination method is of the utmost importance. As well, the impact of self-perceived oral-health related quality of life (OHRQoL) and xerostomia in the quality of life of PD individuals have never been investigated within this purpose.

Our primary aim was to investigate the periodontal status of people with PD. Secondly, we assess the relationship between periodontal clinical measures and clinical characteristics of PD and how quality of life in PD correlates with self-perceived levels of OHRQoL and xerostomia.

## 2. Experimental Section

### 2.1. Study Design

We recruited individuals from the Portuguese Parkinson’s Disease Patient Association (Lisbon branch), between February and March 2020. Inclusion criteria were as follows: people with PD and other parkinsonisms. Exclusion criteria included: unwilling to participate; edentulous; cerebrovascular disease; periodontal treatment during the past six months; and treatment with immunosuppressive chemotherapy. All participants were taking dopaminergic medications, and L-DOPA equivalent doses were calculated for each patient [26].

The study was approved on February 2020 by the Egas Moniz Ethics Committee (Institutional Review Board, protocol 824), and all participants gave informed written consent to the study procedures. We followed the STrengthening the Reporting of OBservational studies in Epidemiology (STROBE) guidelines (Appendix A) [27].

### 2.2. PD Assessment

Motor and non-motor symptoms were assessed using the Movement Disorder Society Unified Parkinson’s Disease Rating Scale (MDS-UPDRS) [28], up to one month prior to the periodontal evaluation. Patient’s motor impairment severity was assessed by the Modified Hoehn and Yahr (H and Y) scale [29]. We categorized H and Y stages as mild (1–2.5) and moderate to severe (3.0–5.0). To assess health-related quality of life (HRQoL), we used the Portuguese version of the eight-item PD Questionnaire (PDQ-8) [30,31]. The PDQ-8 is calculated from eight items representing eight different dimensions. All items are scored on a five-point Likert scale ranging from 0 (“never”) to 4 (“always”). The summed score is divided by total possible score and given in a percentage score out of 100, and higher scores indicate worse HRQoL.

### 2.3. Periodontal Examination and Diagnosis

Full-mouth periodontal examination was performed using a manual periodontal North Carolina probe by two trained and calibrated examiners (VM and JB) (Hu-Friedy; Chicago, IL, USA).

The following parameters were circumferentially measured at six sites per tooth (mesiobuccal, buccal, distobuccal, mesiolingual, lingual and distolingual) in all teeth (except third molars, implants and retained roots): plaque index (PI) [32], gingival recession (REC), periodontal pocket depth (PPD) and bleeding on probing (BoP). PPD referred to the distance from the free gingival margin to the bottom of the pocket. REC was the distance from the cementoenamel junction (CEJ) to the free gingival margin and this assessment was assigned a negative sign if the gingival margin was located coronally to the CEJ. CAL was the algebraic sum of REC and PD measurements for each site. The measurements were rounded to the lowest whole millimetre (mm). Tooth mobility was further appraised [33].

Intra-class correlation coefficient (ICC) values were 0.98 and 0.99, for CAL and PPD, respectively. The intra-examiner ICC ranged from 0.97 to 0.99, for both PPD and CAL.

Periodontitis cases were defined if: interdental CAL ≥ 2 non-adjacent teeth, or Buccal or Oral CAL ≥ 3 mm with PD > 3mm is detectable at ≥ 2 teeth [34]. Then, periodontitis staging was defined according to the 2018 World Consensus. Staging was defined as:CAL at site of greatest loss of 1–2 mm—Stage 1 or mild;CAL at site of greatest loss of 3–4 mm—Stage 2 or moderate;and, CAL at site of greatest loss of ≥ 5 mm—Stage 3/4 or severe/advanced.

### 2.4. Sociodemographic and Oral Health Covariates

By means of a structured questionnaire, we collected information regarding: (1) gender, age, marital status, educational level, occupation; (2) smoking habits; (3) oral hygiene-related behaviors (toothbrush type, toothbrushing frequency, and interproximal cleaning); (5) attitudes and awareness towards oral health; and (6) diabetes mellitus (DM).

We categorically registered Education levels according to the 2011 International Standard Classification of Education (ISCED-2011): elementary (ISCED 1–2 levels), middle (ISCED 3–4 levels), higher (ISCED 5–8 levels). Smoking status was defined as non-smoker (category 0), former smoker (category 1); or active smoker (category 2). DM was based on insulin regimen and/or oral hypoglycaemic medications and was confirmed through haemoglobin A1c (HbA1c).

### 2.5. OHRQoL and Xerostomia Self-Perception Questionnaires

Before the periodontal examination, we used the Portuguese versions of the Oral Health Impact Profile-14 (OHIP-14-PT) [35] and the Summated Xerostomia Inventory (SXI-5) to assess OHRQoL and dry mouth symptoms, respectively.

We measured OHRQoL through the Portuguese version of the Oral Health Impact Profile-14 (OHIP-14-PT) [35], a 14 questions tool with seven domains (functional limitation, physical pain, psychological discomfort, physical disability, psychological disability, social disability and handicap) of OHRQoL. Each question is scored categorically (0 = never, 1 = hardly ever, 2 = occasionally, 3 = fairly often and 4 = very often) [36]. A higher score indicates poorer OHRQoL.

Dry mouth perception was evaluated through the Portuguese version of the Shortening the Xerostomia Inventory (SXI-5), a five-question tool where each question is scored by 0 = never, 1 = occasionally and 2 = frequently. The scores from the five questions are summed, with the result representing the degree of xerostomia the subject feels [37].

### 2.6. Statistical Analysis

The total scores of PDQ-8, OHIP-14 and SXI-5 were calculated and their correspondent descriptive measures, mean and standard deviation (SD), were computed. For analysis purposes, these scores were considered as continuous variables. The data analyses were conducted for all participants and for sample subsets, according to the patient’s motor impairment severity, given by the Modified H and Y scale. The Mann–Whitney test was used to compare the periodontal clinical measures between these subgroups. Spearman’s rank-order correlation coefficient (rho) was used to analyze the correlation among questionnaires scores (MDS-UPDRS, PDQ-8, OHIP-14 and SXI-5) and between these and the periodontal clinical variables. Data were analysed using IBM SPSS Statistics, v. 25, (Armonk, New York, NY, USA). A level of significance of 5% was considered in all inferential analyses.

## 3. Results

### 3.1. Sample Description

From a total of 33 individuals with PD, five participants were excluded because they were edentulous. A final sample of 28 participants were enrolled, with a mean age of 72.3 (±8.1) years, meeting the required inclusion criteria (Table 1). The group was composed mostly by men (82.1%), with idiopathic PD (82.1%) and diagnosed with moderate to severe patient’s motor impairment (H and Y) (64.3%). The prevalence of periodontitis was high (75.0%), and the majority were severe cases (stage III) (46.4%). On average, the participants had 12 teeth missing, and one tooth with pathological mobility. The average percentage of plaque and gum inflammation in the whole mouth were 37.0% (±29.4) and 19.3% (±21.1), respectively. The majority of patients report the use of a manual toothbrush (75.0%) and a last dental visit within the last 6 months (64.3%).

### 3.2. Relationship between PD Staging and Periodontal Status

We found no differences between PD stages (Table 2). Mild PD presented lower prevalence of severe periodontitis than more advanced PD stages (Table 3). An increased number of missing teeth was associated with a worsening of speech and eating tasks (Table 4). Further, worse periodontal measures (PI, BoP and mean PPD) were correlated with deteriorated levels of rigidity and kinetic tremors of the upper extremities (Table 4). Likewise, a more depressive state and hands postural tremors also were correlated with worse levels of BoP.

### 3.3. PD Quality Of Life Impact on Ohrqol and Xerostomia Self-Perception

Spearman’s rank-order correlation coefficient (rho) was used to assess the correlation between total and each domain of PDQ-8 and OHIP-14 (Table 5). All significant correlations were positive, that is the worsening of a domain is associated with worse levels in the other domain. Worse social support was correlated with worse overall OHIP-14 (rho = 0.459, *p* < 0.05), psychological discomfort (rho = 0.518, *p* < 0.01) and psychological disability (rho = 0.534, *p* < 0.01). The deterioration of cognition was correlated with worse psychological discomfort (rho = 0.401, *p* < 0.05) and handicap levels (rho = 0.416, *p* < 0.05) in OHIP-14. Likewise, worse levels of mobility and activities of daily living were correlated with worse levels of psychological disability (rho = 0.445, *p* < 0.05) and handicap (rho = 0.431, *p* < 0.05), respectively.

The correlation between PDQ-8 and SXI-5 scores, considering total and respective domains/items, was also investigated (Table 5). Higher difficulties in eating dry foods were significantly correlated with worse overall quality of life (rho = 0.426, *p* < 0.05) and bodily discomfort (rho = 0.450, *p* < 0.05). Additionally, self-perceived deteriorated cognition was related to difficulties swallowing certain foods (rho = 0.460, *p* < 0.05).

## 4. Discussion

To the best of our knowledge, this study is the first to consistently appraise the periodontal status and clinical measures of interest in a group of PD individuals. Our results show that periodontitis and gum inflammation were highly prevalent in this group of people with PD, and about a third of the population had moderate and severe forms of periodontitis.

This prevalence might be explained by the age, the number of males and the smoking habits in the included sample, recognized risk factors for periodontitis based on a recent representative study in this region [19]. In other words, because it is a very aged sample the existence of gum disease may be more increased. Likewise, men have more prevalence of periodontitis than women, and active and former smokers have much more risk to its development (OR = 3.76) [19]. This prevalence is in line with previous studies, though they have used unsuited periodontal clinical methods [20,21,22,23]. Additionally, the prevalence of periodontitis in this age-group is in agreement with previous studies developed in this region, where these age groups have high levels of periodontal disease [19,38,39].

Nevertheless, there are a number of other important characteristics that may elucidate the increased gum inflammation observed, which stands out the hygiene habits. This particular population has good toothbrushing frequency (with a regimen of twice or more per day) and regular dental visits, however the majority does not perform interdental hygiene, and this fact increases the likelihood of gum inflammation [40]. Importantly, we anticipate that interdental cleaning is a major challenge for PD individuals due to the characteristics of the disease itself (fine motor issues, cognitions deficits), thus home interdental cleaning (such as oral irrigation) and electric toothbrushes should be recommended to bypass common fine motor difficulties [41].

Interestingly, the comparison of PD staging through the modified H and Y scale did not find any marked difference between the groups, but the sample was limited and may represent a restrictive characteristic. Using the MDS-UPDRS, deteriorated levels of kinetics tremor, postural tremor and rigidity of the upper extremities were associated with increased plaque accumulation, gum bleeding and, as consequence, deepest periodontal pockets. Comprehensively, the progression of PD may ultimately result in impaired oral hygiene habits, and this is the first study to prove the likelihood of this association. Additionally, severe stages of PD had more teeth missing, and our data confirmed that the more teeth lost the worse perception of speech and eating functions in PD. The number of missing teeth was a relevant measure in this analysis, since severe stages of PD had less teeth presence, though this difference was not significant. Our data show that the more teeth lost, the worse the speech and eating ability in PD.

Our findings also corroborate the premise that perceived quality of life in PD might be interconnected with OHRQoL and xerostomia. On the one hand, psychological domains (psychological discomfort and disability) of OHIP-14 and social support from PDQ-8 had the strongest relation, and further investigations should better explore this interaction. Regarding self-perceived dry mouth, more difficulties eating dry foods influenced PD quality of life and caused more bodily discomfort, while worse cognition levels were associated with difficulties swallowing certain foods. The presence of swallowing impairment in PD individuals is not new and has been previously reported [20]. Others, PD patients have sialorrhea as a characteristic symptom, although dry mouth in this type of older population is frequent due to polymedication [19]. Additionally, xerostomia is poorly rated on the MDS-UPDRS scale and, therefore, the SXI may be a complement in PD clinical follow-up. All in all, the self-reported instruments used are valid for future studies involving PD and oral health.

The interplay between periodontitis and systemic inflammation is well documented [42,43]. A recent systematic review confirmed that periodontal bacterial load and gum inflammatory burden can intensify the neuroinflammation in the central nervous system in Alzheimer’s disease (AD), favoring the onset and progression of this neurodegenerative condition [44]. Accumulating evidence has shed light on how neuroinflammation can play a key role on the PD pathogenesis and neurodegeneration [45,46]. As a potential contributor to neuroinflammation, periodontitis can cause a subclinical inflamed state in PD patients and may negatively alter the neuronal environment as in AD as previously proposed [25,47,48], however, this is merely speculative at this stage and shall be investigated in the future.

The present study had some limitations. The small size of the group is the main shortcoming, and for that reason the results should be interpreted with caution, as this limits the validity of these results and warrants future confirmation with prospective studies, since there are inherent biases in cross-sectional studies, such as selection bias. However, these studies comprise two months of inclusion and follows the rigorous STROBE guideline. This study did not ascertain the technique and the brushing efficiency of the participants along with the motor assessment in PD, a factor that should be considered in future studies. Second, we did not consider the patient’s dominant arm during tooth brushing, which led us to count the measures of the upper limbs as the average. In the future, studies shall consider the influence of PD progression on the dominant arm of the patient’s adaptive behaviour, for example changing arms during oral hygiene.

This is the first study to use appropriate periodontal assessment methods. The measures of interest were assessed by trained and calibrated examiners and the most up-to-date definitions of PD and periodontitis were used [47], making these results current and of high scientific interest. As an observational study, this investigation is unable to estimate a causality between PD and gum disease as well the impact of particular clinical features of PD on the progression of periodontitis and vice versa.

## 5. Conclusions

Periodontitis was highly prevalent in this group of people with PD. Deteriorated levels of the upper extremities in advanced stages of PD influence the periodontal status and hygiene habits. Quality of life in PD appears to be associated with self-perceived OHRQoL and xerostomia. Future studies should consider the impact of periodontitis on the quality of life of PD and the potential inflammatory burden of periodontitis.

## Figures and Tables

**Table 1 medicina-56-00383-t001:** Participant characteristics.

Variable	Result
Age, mean (SD) (years)	72.3 (± 8.1)
Age Range (min-max) (years)	57–92
Gender, *n* (%)	
Female	5 (17.9)
Male	23 (82.1)
Education, *n* (%)	
Elementary	6 (21.4)
Middle	12 (42.9)
Higher	10 (35.7)
Hoehn & Yahr Scale, *n* (%)	
Mild (1 to 2.5)	10 (35.7)
Moderate to Severe (3 to 5)	18 (64.3)
Parkinson’s Disease, *n* (%)	
Idiopathic	23 (82.1)
Atypical	5 (17.9)
Periodontal status, *n* (%)	
Healthy	7 (25.0)
Periodontitis	21 (75.0)
Stage 1—Mild	2 (7.1)
Stage 2—Moderate	6 (21.4)
Stage 3—Severe	13 (46.4)
Teeth with mobility, mean (SD)	1 (2)
Missing teeth, mean (SD)	12 (7)
Plaque Index, mean (SD) (%)	37.0 (29.4)
BoP, mean (SD) (%)	19.3 (21.1)
Mean PPD, mean (SD) (mm)	2.1 (0.8)
Mean CAL, mean (SD) (mm)	3.2 (1.8)
Mean REC, mean (SD) (mm)	1.2 (1.2)
Toothbrush type, *n* (%)	
Manual	21 (75.0)
Electric	7 (25.0)
Last dental visit, *n* (%)	
< 6 months	18 (64.3)
6–12 months	4 (14.3)
> 12 months	6 (21.4)
Toothbrushing	
Once a day	9 (32.1)
Twice or more a day	19 (67.9)
Interproximal cleaning	
Never	9 (32.1)
No	8 (28.6)
Often/Yes	11 (39.3)
Smoking habits, *n* (%)	
Never	16 (57.1)
Former smoker	7 (25.0)
Active smoker	5 (17.9)
Diabetes Mellitus, *n* (%)	**3 (10.7)**

BoP—Bleeding on Probing; CAL—Clinical Attachment Loss; PD—Probing Depth; REC—Recession; SD—Standard Deviation.

**Table 2 medicina-56-00383-t002:** Comparison of age, PDQ-8, OHIP-14, SXI-5 total scores and periodontal clinical measures according to PD progression, based on the Hoehn and Yahr (H and Y) scale.

Variable	PD Progression		*P*-Value^#^
	Mild (*n* = 10)	Moderate to Severe (*n* = 18)	
Variable, mean (SD)	70.4 (5.7)	73.3 (9.1)	0.382
Age (years)	5.9 (3.0)	10.6 (7.2)	0.191
PDQ-8 (total score)	7.1 (7.2)	11.0 (12.6)	0.436
OHIP-14 (total score)	7.8 (1.9)	7.7 (2.2)	0.759
SXI-5 (total score)	9.5 (7.9)	13 (7.1)	0.191
No. of missing teeth	31.6 (23.4)	39.9 (32.6)	0.524
Plaque Index (%)	23.9 (32.4)	16.8 (11.5)	0.796
BoP (%)	2.2 (1.2)	2.0 (0.5)	0.382
Mean PPD (mm)	3.3 (2.3)	3.2 (1.6)	0.356
Mean CAL (mm)	1.1 (1.2)	1.2 (1.2)	0.724
Mean REC (mm)	70.4 (5.7)	73.3 (9.1)	0.382

^#^ Mann-Whitney test, *p* < 0.05.

**Table 3 medicina-56-00383-t003:** Periodontal status according to PD progression.

Periodontal Status, *n* (%)	PD Progression	
	Mild (*n* = 10)	Moderate to Severe (*n* = 18)
Healthy	3 (30.0)	4 (22.2)
Mild Periodontitis	2 (20.0)	0 (0.0)
Moderate Periodontitis	1 (10.0)	5 (27.8)
Severe Periodontitis	4 (40.0)	9 (50.0)

**Table 4 medicina-56-00383-t004:** Correlation between MDS-UPDRS items and periodontal clinical measures.

MDS-UPDRS	No. of Missing Teeth	Plaque Index (%)	BoP (%)	Mean PPD (mm)	Mean CAL (mm)	Mean REC (mm)
1.1 (Cognitive Impairment)	0.072	−0.152	−0.199	−0.321	−0.117	−0.071
1.3 (Depressed mood)	0.234	0.069	0.394 *	0.132	0.280	0.263
1.5 (Apathy)	0.350	−0.066	−0.040	−0.163	0.098	0.203
2.1 (Speech)	0.419 *	0.005	0.101	−0.098	0.044	−0.001
2.2 (Saliva and drooling)	−0.044	0.060	0.175	−0.037	−0.202	−0.256
2.3 (Chewing and swallowing)	0.319	0.137	0.126	0.023	0.225	0.178
2.4 (Eating tasks)	0.404 *	−0.141	0.019	−0.098	0.076	0.046
2.6 (Hygiene)	−0.003	−0.127	0.002	−0.121	−0.013	−0.096
2.7 (Writing)	0.185	−0.055	−0.014	−0.040	0.062	−0.004
2.10 (Tremor)	0.004	0.220	0.269	0.266	0.544	−0.111
3.3. (Rigidity UE)	0.180	0.387 *	0.382 *	0.452 *	0.230	0.004
3.4 (Finger tapping)	0.252	0.145	0.105	0.146	0.288	0.200
3.5 (Hand movements)	0.059	0.080	0.049	0.136	0.049	−0.083
3.6 (Pronation)	0.127	0.013	−0.127	−0.027	0.062	−0.009
3.15 (Hands postural tremor)	0.102	0.060	0.451 *	0.235	0.212	0.120
3.16 (Hands kinetic tremor)	−0.100	0.559 **	0.541 **	0.431 *	0.213	0.018
3.17 (Rest tremor amplitude UE)	−0.134	0.188	0.057	0.258	0.119	0.061
3.17e (Rest tremor amplitude Lip/jaw)	0.263	0.012	0.143	0.048	0.107	0.131
3.18 (Constancy of rest tremor)	−0208	0.128	0.100	0.208	−0.038	−0.123

Spearman correlation, * *p* < 0.05, ** *p* < 0.01.

**Table 5 medicina-56-00383-t005:** Correlation between Parkinson’s Disease Questionnaire (PDQ-8) with Oral Health Impact Profile (OHIP-14) and Summated Xerostomia Inventory-5 (SXI-5) scores, total and per domain.

Variable	PDQ−8 Total	Mobility	ADLs	Emotional Well-Being	Stigma	Social Support	Cognition	Communication	Bodily Discomfort
**OHIP−14** **Total**	0.278	0.192	0.179	0.000	−0.004	0.459*	0.342	0.116	0.031
Functional Limitation	0.243	0.171	0.301	−0.064	−0.035	0.208	0.361	0.148	−0.126
Psychological Pain	0.096	0.196	−0.035	−0.064	−0.062	0.120	0.183	0.210	−0.076
Psychological Discomfort	0.283	0.205	0.123	0.275	0.265	0.518 **	0.401 *	−0.022	0.034
Physical Disability	0.260	0.197	0.350	0.196	0.111	0.271	0.307	−0.060	0.069
Psychological Disability	0.322	0.445 *	0.119	0.004	0.034	0.534 **	0.194	0.154	0.245
Social Disability	0.291	0.231	0.291	−0.024	0.023	0.320	0.254	0.272	0.197
Handicap	0.317	0.222	0.431*	0.223	0.223	0.007	0.416 *	0.152	0.208
**SXI−5 Total**	0.291	0.183	−0.037	0.068	0.257	0.197	0.318	0.076	0.290
Dry mouth	0.121	0.180	−0.031	0.055	0.163	0.127	0.206	−0.034	0.129
Difficulty eating dry foods	0.426 *	0.328	0.004	0.069	0.255	0.224	0.270	0.220	0.450 *
Dry mouth when eating meal	−0.093	−0.056	0.025	−0.042	−0.101	0.159	0.109	−0.067	−0.199
Difficulties swallowing certain foods	0.208	0.087	0.087	0.344	0.149	0.060	0.460 *	−0.039	0.234
Dry lips	0.209	0.150	−0.030	−0.201	0.229	0.186	0.152	0.062	0.168

ADLs—Activities of Daily Living. Spearman correlation, * *p* < 0.05, ** *p* < 0.01.

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
