# Peer review of "Parkinson’s Disease, Periodontitis and Patient-Related Outcomes: A Cross-Sectional Study"

_medicina, 2020, doi:10.3390/medicina56080383_

Round 1

Reviewer 1 Report

Dear authors,
congratulations for your efforts and your work during a difficult time for the whole world.

Please kindly consider the following revisions for your manuscript:

1) To classify periodontal patients with the new classification you are required to look at the interdental CAL of your patients, please revise accordingly or redo analyses if needed

2) Please consider discussing the prevalence of periodontitis in Portugal in this particular age-group

3) Please revise patient characteristics and give them categorised per diseases severity after correct categorisation of the patients with the new classification (see comment #1)

Regards

Author Response

We are pleased with the opportunity to revise and resubmit our manuscript “Parkinson’s Disease, Periodontitis and patient-related outcomes: a cross-sectional study” (Manuscript ID medicina-849606).

We are very grateful for the editor and reviewers’ comments, all have been considered and taken into profound consideration.

Manuscript changes are highlighted in the revised manuscript. Our point-by-point responses to all comments are detailed below. We hope the revised manuscript will better suit the Journal of Personalized Medicine. We are happy to consider further revisions and we thank you for your continued interest in our research.

REVIEWER 1:

Dear authors,

congratulations for your efforts and your work during a difficult time for the whole world.

Please kindly consider the following revisions for your manuscript:

1) To classify periodontal patients with the new classification you are required to look at the interdental CAL of your patients, please revise accordingly or redo analyses if needed

Answer: We have revised the description, as we overlooked the interdental CAL of our patients and followed strictly this classification. We rephrased to: “Periodontitis cases were defined if: interdental CAL ≥2 non‐adjacent teeth, or Buccal or Oral CAL ≥3 mm with PD >3 mm is detectable at ≥2 teeth [32].” (Page 3, Lines 109-111).

2) Please consider discussing the prevalence of periodontitis in Portugal in this particular age-group

Answer: We discussed the prevalence of periodontitis in Portugal in this age-group that is, in fact, in agreement with literature: “Additionally, the prevalence of periodontitis in this age-group is in line with previous studies developed in this region, where this age groups have high levels of periodontal disease [46–48].” (Page 7, Lines 214-215).

3) Please revise patient characteristics and give them categorised per diseases severity after correct categorisation of the patients with the new classification (see comment #1)

Answer: We have answered to comment #1, however, since the classification was made accordingly and appropriately we have categorized according to the new classification. We appreciate this remark.

Reviewer 2 Report

The authors have submitted a manuscript describing a cross sectional study to understand periodontal disease in Parkinson's disease patients. They have also reported on the correlation of quality of life in Parkinson’s disease on the oral health-related quality of life (OHRQoL) and self-perceived xerostomia. At the outset, the authors have conducted and presented this study well. It is well known that Parkinson’s disease patients have difficulty in maintaining good oral hygiene due to physical restrictions and hence this data confirms this. Please consider the following observations for improving this manuscript.

  1. Page 3; Lines 110-112: The authors have used the new periodontal classification for diagnosis and staging. However, the usage seems incorrect and incomplete. The definition for staging is interdental clinical attachment loss (CAL) and it is not clear if the authors used this or CAL at any other site.
  2. Similarly, the authors have not indicated Stage IV in the staging of periodontal disease. The mean number of missing teeth is 12 (Table 1). This will be considered Stage IV if 5 or more of these teeth were lost due to periodontal disease and patients typically have other issues such as bite collapse, ridge defects, etc.
  3. The authors also have not presented the grading which must be an integral part of staging and grading of periodontitis according to the new classification.
  4. The authors use PD as the abbreviation for Parkinson’s disease and PPD for periodontal probing depth. In some areas like in tables, they use PD for periodontal probing depths conflicting with PD Parkinson’s Disease. Please use dissimilar abbreviations for less confusion while reading. Also, many other authors use PD for periodontal disease to add more difficulty in reading this manuscript.
  5. The study only has 28 patients. The authors are requested to expand on the limitation of this in terms of validity of the results. Informing readers about inherent biases in a cross-sectional study such as selection bias and their effect on obtained results would improve this manuscript.
  6. The detailed data and presentation of the Parkinson’s disease rating, severity, OHRQoL, and xerostomia is commendable.

Author Response

We are pleased with the opportunity to revise and resubmit our manuscript “Parkinson’s Disease, Periodontitis and patient-related outcomes: a cross-sectional study” (Manuscript ID medicina-849606).

We are very grateful for the editor and reviewers’ comments, all have been considered and taken into profound consideration.

Manuscript changes are highlighted in the revised manuscript. Our point-by-point responses to all comments are detailed below. We hope the revised manuscript will better suit the Journal of Personalized Medicine. We are happy to consider further revisions and we thank you for your continued interest in our research.

REVIEWER 2:

The authors have submitted a manuscript describing a cross sectional study to understand periodontal disease in Parkinson's disease patients. They have also reported on the correlation of quality of life in Parkinson’s disease on the oral health-related quality of life (OHRQoL) and self-perceived xerostomia. At the outset, the authors have conducted and presented this study well. It is well known that Parkinson’s disease patients have difficulty in maintaining good oral hygiene due to physical restrictions and hence this data confirms this. Please consider the following observations for improving this manuscript.

Page 3; Lines 110-112: The authors have used the new periodontal classification for diagnosis and staging. However, the usage seems incorrect and incomplete. The definition for staging is interdental clinical attachment loss (CAL) and it is not clear if the authors used this or CAL at any other site.

Answer: We have revised the description, as we overlooked the interdental CAL of our patients and followed strictly this classification. We rephrased to: “Periodontitis cases were defined if: interdental CAL ≥2 non‐adjacent teeth, or Buccal or Oral CAL ≥3 mm with PD >3 mm is detectable at ≥2 teeth [32].” (Page 3, Lines 109-111).

Similarly, the authors have not indicated Stage IV in the staging of periodontal disease. The mean number of missing teeth is 12 (Table 1). This will be considered Stage IV if 5 or more of these teeth were lost due to periodontal disease and patients typically have other issues such as bite collapse, ridge defects, etc.

Answer:  We corrected the definition of staging. Since stage 4 is considered when 5 or more of these teeth were lost due to periodontal disease, and part of these patients may have cognitive decline and may introduce bias due to misinformation, we decided to base our staging in clinical information rather than in past history. Also, previous studies have also performed this type of grouping (Botelho et al. 2019, Machado et al. 2019, Botelho et al. 2020).

References:

Botelho J, Machado V, Proença L, et al. Study of Periodontal Health in Almada-Seixal (SoPHiAS): a cross-sectional study in the Lisbon Metropolitan Area. Sci Rep. 2019;9(1):15538. Published 2019 Oct 29. doi:10.1038/s41598-019-52116-6

Machado V, Botelho J, Ramos C, et al. Psychometric properties of the Brief Illness Perception Questionnaire (Brief-IPQ) in periodontal diseases. J Clin Periodontol. 2019;46(12):1183-1191. doi:10.1111/jcpe.13186

Botelho J, Machado V, Proença L, et al. Perceived xerostomia, stress and periodontal status impact on elderly oral health-related quality of life: findings from a cross-sectional survey. BMC Oral Health. 2020;20(1):199. Published 2020 Jul 10. doi:10.1186/s12903-020-01183-7

The authors also have not presented the grading which must be an integral part of staging and grading of periodontitis according to the new classification.

Answer: We decided to not use the grading because, as stated in Tonetti et al. 2018, and quote, “Grade should be used as an indicator of the rate of periodontitis progression”. In our interpretation, progression demands a longitudinal and progressive observation which is not the case of our observational study. As the premise was not fulfilled we opted to minimize bias of analysis and not providing grading. However, if the reviewer and editor find it relevant that this information should be present, we are available to add it, in the reader’s best interest.

The authors use PD as the abbreviation for Parkinson’s disease and PPD for periodontal probing depth. In some areas like in tables, they use PD for periodontal probing depths conflicting with PD Parkinson’s Disease. Please use dissimilar abbreviations for less confusion while reading. Also, many other authors use PD for periodontal disease to add more difficulty in reading this manuscript.

Answer: We apologize for these typos. We have corrected them accordingly through the manuscript.

The study only has 28 patients. The authors are requested to expand on the limitation of this in terms of validity of the results. Informing readers about inherent biases in a cross-sectional study such as selection bias and their effect on obtained results would improve this manuscript.

Answer: We expanded the possible impact on the validity by stating “Additionally, the prevalence of periodontitis in this age-group is in agreement with previous studies developed in this region, where these age groups have high levels of periodontal disease [19,38,39].” (Page 7, Lines 214-215).

The detailed data and presentation of the Parkinson’s disease rating, severity, OHRQoL, and xerostomia is commendable.

Answer: We appreciate your commentaries, time and effort reviewing our manuscript.

Round 2

Reviewer 1 Report

No further comments, congratulations for your work.